# Application of seasonal-adjusted hybrid models for forecasting Discomfort Index in a heat-prone region of Bangladesh

Amrin Binte Ahmed[1]*, Md. Mahin Uddin Qureshi[1]*, Mohammad Mahboob Hussain Khan[2], Adisha Dulmini[3], Mohammad Ashraful Haque Mollah[4], Rumana Rois[1]*

1 Department of Statistics and Data Science, Jahangirnagar University, Dhaka, Bangladesh, 2 Bangladesh Meteorological Department (BMD), Dhaka, Bangladesh, 3 Department of Business and Law, University of Wollongong, Wollongong, New South Wales, Australia, 4 Action Research Centre for Resilience and Youth, Dhaka, Bangladesh

* amrin.stu2019@juniv.edu (ABA); mdmahin.stu2017@juniv.edu (MMUQ); rois@juniv.edu (RR)

## Abstract

Extreme heat and humidity pose an increasing threat to human health, labor productivity, and overall well-being, particularly in heat-vulnerable and rapidly urbanizing regions such as Rajshahi, Bangladesh. As rising temperatures and elevated humidity levels intensify exposure to heat stress, accurate forecasting has become essential for effective early warning systems and climate-resilient urban planning. However, modeling thermal discomfort is challenging due to the need to analyze long-term, high-frequency meteorological time series data with complex seasonal and nonlinear structures. Therefore, this study applies and evaluates seasonal-adjusted, machine-learning-based hybrid models to forecast thermal discomfort using 40 years (1985–2024) of daily temperature and humidity data. The Thom's Discomfort Index (DI), a standard measure of thermal stress, was calculated and decomposed using the STL (Seasonal-Trend decomposition based on LOESS) method to separate trend, seasonality, and residual components. A total of 128 hybrid model combinations were implemented by integrating traditional time series models (ARIMA, TBATS, ETS, and GARCH) with machine learning techniques (ANN, Prophet, SVR, Decision Trees, Random Forests, XGBoost, LSTM, and GRU). Among all models, the STL-TBATS-LSTM hybrid achieved the best performance, with MAE = 0.4810, MAPE = 2.1230, RMSE = 0.6381, and MASE = 0.6644, followed closely by STL-TBATS-DTR. Historical analysis from 1985 to 2024 revealed strong seasonal peaks in discomfort from June to August, along with a clear long-term increase in both the frequency and intensity of high-discomfort days. Forecasts for 2025–2027 project a substantial rise in thermal stress, with approximately 39.8% of days falling under "High Discomfort" and 1.2% under "Severe Discomfort." These findings highlight the escalating burden of heat stress in Bangladesh and underscore the urgency of STL-based hybrid forecasting

**Data availability statement:** All relevant data are within the paper and its Supporting information files.

**Funding:** The author(s) received no specific funding for this work.

**Competing interests:** The authors have declared that no competing interests exist.

models in supporting climate adaptation strategies and enhancing public health preparedness.

---

## 1. Introduction

Climate and prevailing weather conditions have a direct impact on human comfort [1]. Environmental factors play a key role in regulating the heat exchange of the body with its surroundings and can contribute to the sensations of heat-related stress [2]. Understanding local thermal discomfort conditions is essential due to their strong association with increased morbidity and mortality rates [3–6]. Prolonged exposure to high levels of thermal discomfort can lead to a higher incidence of heat-related health issues and fatalities [7]. Various biometeorological indices quantify thermal discomfort by considering environmental factors like air temperature, relative humidity, and wind speed to assess human heat stress [8–10]. Solar radiation and surface temperatures further affect thermal perception [11]. However, individual experiences of discomfort can vary based on physiological, psychological, and behavioral factors [12].

Among the various indices, Thom's Discomfort Index (DI) has emerged as a widely adopted measure due to its simplicity and dependence on two readily available meteorological variables: air temperature and relative humidity [13]. It has a strong correlation with perceived human discomfort, making it a valuable tool, especially in regions like Bangladesh, where access to comprehensive meteorological data may be limited. Since temperature and humidity data are easily obtainable and exhibit significant variability across time and space, DI serves as a pragmatic choice for large-scale assessments. The index quantifies thermal discomfort by using these parameters and categorizes the results into easily interpretable levels of human comfort or discomfort [14]. This straightforward interpretability enhances DI's utility for public health advisories and early warning systems, particularly in resource-constrained settings.

In recent years, the global impacts of climate change have brought increased attention to the rise in thermally uncomfortable days, particularly in low-income and tropical regions. Zhang et al. (2023) project that low-income regions, particularly those in warmer latitudes, will experience a significant rise in thermally uncomfortable days, intensifying existing vulnerabilities [15]. Ullah et al. (2024) report substantial increases in both the frequency and intensity of heat stress across urban centers in the Arabian Peninsula [16]. In a study of a tropical megacity, Gupta et al. (2024) estimated that nearly 0.9 million individuals were exposed to moderate to severe heat stress, with mid-rise buildings contributing notably to elevated thermal discomfort [17]. Interestingly, Golshan et al. (2021) found that students across various global locations experienced higher levels of thermal discomfort during the winter months, highlighting the complex interplay between climate, behavior, and infrastructure [18]. Together, these findings underscore the multifaceted nature of thermal discomfort and the urgency for implementing adaptive urban strategies, such as green pavements, water spray cooling, and heat-reflective infrastructure [15,17].

Bangladesh, with its hot and humid climate, is already experiencing marked increases in thermal discomfort. Ekra et al. (2024) reported significant increases in Heat Discomfort Index (DI), Humidex, and wet-bulb temperature across the country, with coastal regions experiencing more severe changes [19]. Talukdar et al. (2017) found that over 50% of the population in Mymensingh experienced discomfort from April to October [20]. Sharmin et al. (2020) modeled outdoor thermal sensation in Dhaka using both meteorological and personal factors [21]. Rahman et al. (2021) projected a continuous rise in heat index (HI) for the mid-century, with northern, northeastern, and south-central regions expected to face more significant increases [22]. These region-specific findings highlight the urgent need for localized adaptation strategies, such as promoting green urban spaces, implementing cool roof technologies, and improving public awareness around heat-related risks [23,24].

To meet the increasing demand for accurate forecasting, recent research has concentrated on hybrid models that merge traditional statistical methods with machine-learning algorithms to enhance predictive precision. Susanti et al. (2017) developed a hybrid model integrating time series regression, autoregressive integrated moving average (ARIMA), and artificial neural networks (ANN) to forecast data with trend, seasonal, and calendar variation patterns [25]. In the same way, Yunis et al. (2024) proposed hybrid models combining support vector regression (SVR), seasonal autoregressive integrated moving average (SARIMA), Long Short-Term Memory (LSTM), and Prophet for air pollution prediction, with LSTM-Prophet achieving the highest accuracy [26]. Thakur et al. (2021) surveyed hybrid models used in hydrological time series forecasting, highlighting the benefits of combining neural networks with conventional methods [27]. Sulandari et al. (2020) introduced singular spectrum analysis (SSA)-based hybrid models, two-level seasonal neural network (TLSNN) and two-level complex seasonal neural network (TLCSNN), which incorporate flexible trend functions, harmonics, and neural networks to improve forecasting performance for complex time series patterns [28]. These studies consistently demonstrated that hybrid models tend to outperform individual forecasting methods in terms of accuracy and flexibility.

Expanding upon these worldwide progressions, researchers in Bangladesh have recently started implementing hybrid models in diverse domains. Qureshi et al. (2025) introduced two novel algorithms for developing machine learning (ML)-based hybrid models, including seasonal-adjusted variants, to forecast heatwave warnings. These models are designed to manage high frequency datasets and effectively capture complex seasonal patterns [29]. Saif et al. (2024) compared ARIMA, neural networks, and a hybrid model to forecast fish production and concluded that the hybrid approach yielded the highest accuracy [30]. In the context of flood prediction, Dipro et al. (2023) evaluated ARIMA, SARIMAX, and Prophet models using inputs such as temperature, rainfall, and river water-level data [31]. Similarly, Ferdoush et al. (2021) demonstrated that a hybrid machine learning model combining random forest regression (RFR) and bidirectional LSTM outperformed standard LSTM models in short-term electricity load forecasting in Bangladesh [32]. In another study focused on precipitation forecasting in northern Bangladesh, a hybrid model combining M5 Prime (M5P) model tree and SVR achieved high predictive accuracy, with $R^2$ values reaching 0.87 and 0.92 for Rangpur and Sylhet, respectively, outperforming the individual models [33].

Despite this progress, limited research has explored the forecasting of thermal discomfort using seasonally adjusted hybrid models in the heat-vulnerable regions of Bangladesh, such as Rajshahi. Although DI is a widely used biometeorological indicator, it remains underutilized in long-term forecasting efforts aimed at informing public health and urban planning. Conventional models often struggle to fully capture the complex, multi-scale patterns in climatic data, which include strong seasonality, non-stationary trends, and non-linear residuals—a limitation that can be critical in a rapidly warming region like Bangladesh. To address this critical gap, the present study introduces a novel forecasting approach by applying the seasonal-adjusted hybrid algorithms of Qureshi et al. (2025) [29]. The novelty of our method lies in the strategic integration of seasonal-trend decomposition using Loess (STL) for effective seasonal decomposition, an advanced time series model (like Trigonometric Seasonality, Box-Cox Transformation, ARMA Errors, Trend, and Seasonal Components (TBATS)) for managing complex and multiple seasonal patterns, and an ML model (like LSTM) to capture nonlinear temporal dynamics. This integrated hybrid model surpasses conventional methods by delivering enhanced accuracy in

 

forecasting thermal discomfort, specifically tailored to Bangladesh's highly variable climate. The resulting high-precision projections are not only a technical advancement but also play a crucial role in enabling proactive public health interventions, optimizing early warning systems, and guiding data-driven urban planning. Such improvements are essential for strengthening climate resilience among vulnerable populations in heat-prone regions like Rajshahi.

## 2. Materials and methods

### 2.1. Study area and data collection

This study focuses on Rajshahi, a northwestern district of Bangladesh characterized by a hot and dry climate, particularly during the pre-monsoon and summer seasons [34]. Daily meteorological data, including average temperature and relative humidity, were obtained from the Bangladesh Meteorological Department (BMD) for the period from January 1, 1985, to December 31, 2024. Fig 1 presents the spatial distribution of the average annual number of severe discomfort days across 34 meteorological stations in Bangladesh, providing an overview of historical thermal stress patterns. To ensure data integrity and continuity, all missing values in the meteorological variables were addressed using time series-based imputation techniques before analysis. These variables were then utilized to calculate the DI, which is a well-known composite thermal indicator for evaluating human thermal stress. The resulting DI time series laid the groundwork for all the subsequent forecasting analyses. All data processing, modeling, and visualization tasks were performed using the R programming language.

### 2.2. Discomfort Index (DI) definition and classification

The DI is a widely used empirical indicator that estimates the level of thermal discomfort experienced by individuals due to the combined effects of temperature and humidity [35]. It plays a crucial role in evaluating human comfort, particularly in the context of climate monitoring, public health, and energy management [36]. In this study, Thom's Discomfort Index was used, which is computed using the following formula [14]:

$$DI = T - \{0.55 \cdot (1 - 0.01 \cdot RH) \cdot (T - 14.5)\}. \tag{1}$$

where DI is the Discomfort Index, T is the Average Air Temperature (°C), and RH is the Relative Humidity (%). Based on the calculated DI values, thermal stress is classified into six levels, ranging from no discomfort to extreme heat stress, as presented in Table 1. These categories help in interpreting the severity of heat stress and its potential impact on public health.

### 2.3. Hybrid model's forecasting framework

The hybrid modeling strategies used in this study are based on the earlier work [29]. To evaluate model performance, the DI time series was partitioned into 75% training data and 25% test data. As illustrated in Fig 2, the first approach applies a time series (TS) or ML model to the original data, followed by modeling the residuals using a complementary method. The final forecast is obtained by combining the primary forecast and the forecasted residuals. The second approach incorporates a seasonal adjustment step using STL decomposition to isolate the seasonal component from the original DI data. A model was then fitted to the seasonal-adjusted data and the residuals were further modeled. The final forecast was generated by aggregating the seasonal component with both the forecasted components from the adjusted model and its residuals. These workflows allow for better handling of trend, seasonality, and nonlinearity in the DI time series.

### 2.4. Forecasting time series and machine learning models

We utilized a diverse set of forecasting models, including classical time-series, machine learning, ensemble, and deep learning models. These models were selected to capture the linear, nonlinear, seasonal, and volatile characteristics of the

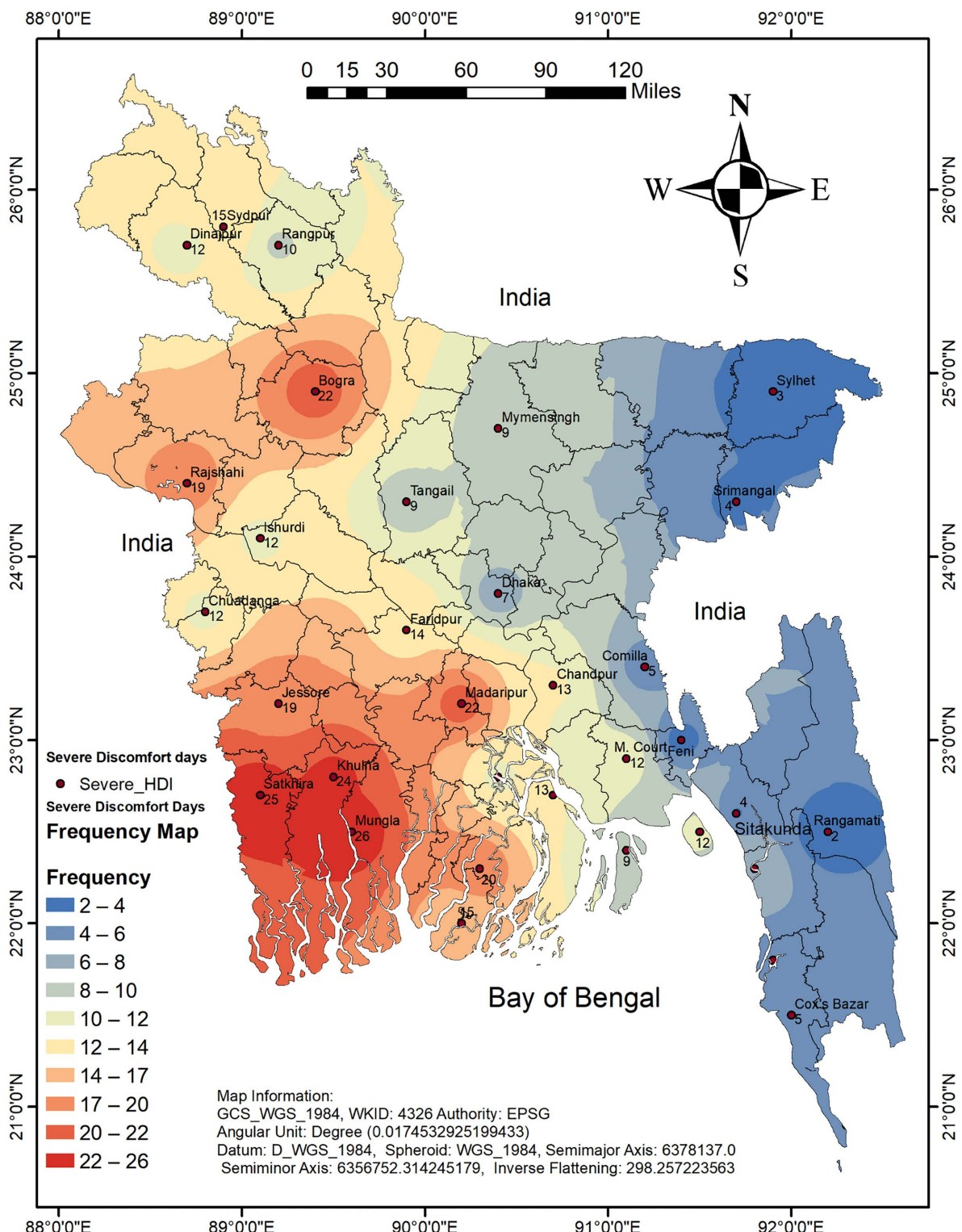

**Fig 1. Spatial distribution of the average annual number of severe discomfort days across 34 meteorological stations in Bangladesh.** The map was generated by the authors using QGIS version 3.30. Administrative boundary data were sourced from the Humanitarian Data Exchange (HDX), provided by OCHA Bangladesh (available at: https://data.humdata.org/dataset/cod-ab-bgd). This is an original figure created for illustrative purposes; it contains no proprietary basemap layers and is compliant with CC BY 4.0 licensing.

**Table 1. Discomfort Index (DI) classification and associated thermal stress levels [13,14].**

| Class | DI Range (°C) | Category | Description |
|---|---|---|---|
| 1 | DI < 21 | No Discomfort | The weather is comfortable with no noticeable heat stress. |
| 2 | 21 ≤ DI < 24 | Mild Discomfort | A small portion of the population may feel slightly uncomfortable. |
| 3 | 24 ≤ DI < 27 | Moderate Discomfort | More than half of the population may feel uncomfortable. |
| 4 | 27 ≤ DI < 29 | High Discomfort | Most people feel hot and uncomfortable. |
| 5 | 29 ≤ DI < 32 | Severe Discomfort | Nearly everyone experiences significant heat stress. |
| 6 | DI ≥ 32 | Extreme Heat Stress | Conditions are dangerous and may pose serious health risks. |

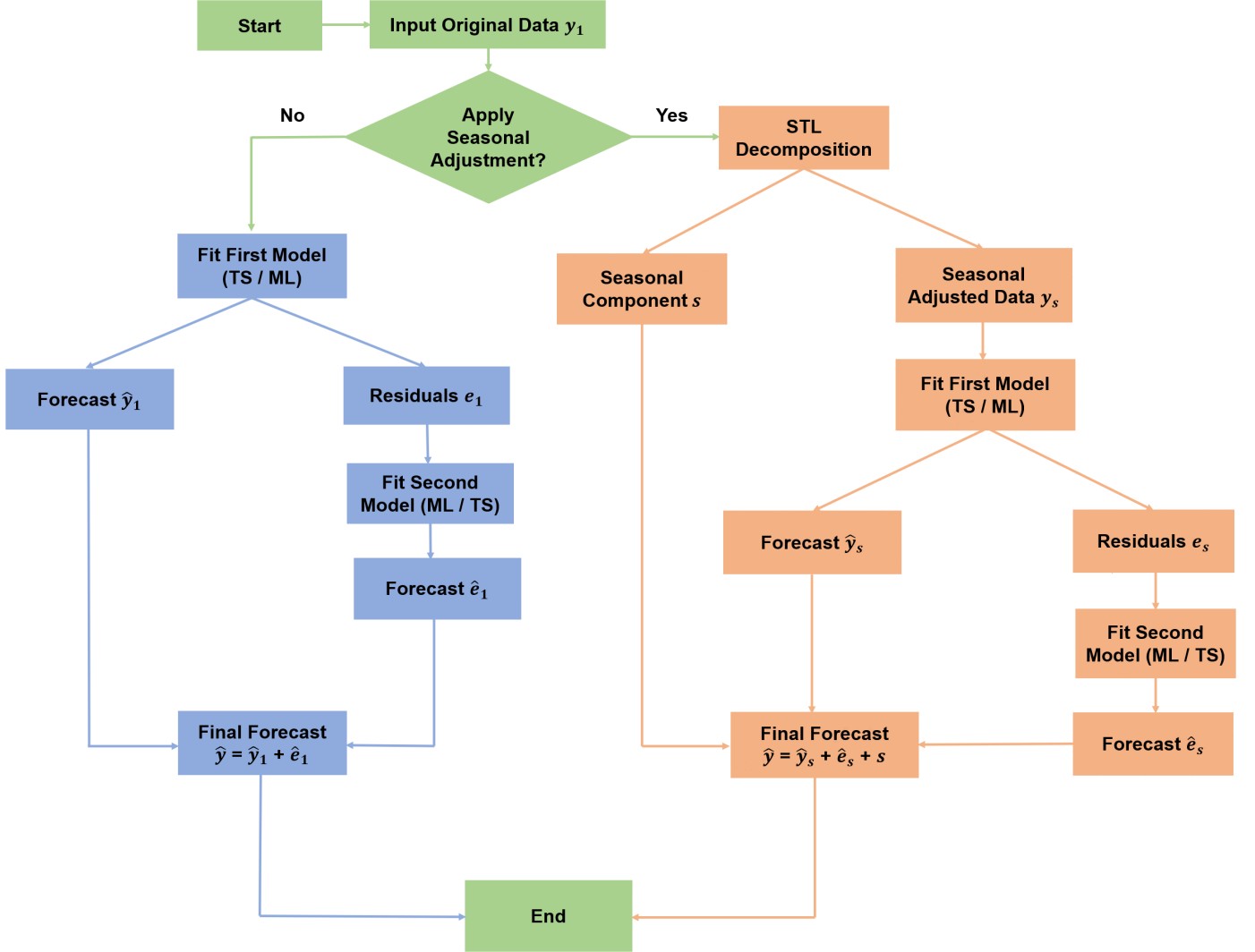

**Fig 2. Workflow of hybrid forecasting models for Discomfort Index (DI) prediction.**

DI time series. Each model was either applied individually or integrated into a hybrid framework to enhance the predictive performance. Detailed descriptions of the models used are provided in the following subsections (2.4.1–2.4.12).

**2.4.1. Autoregressive integrated moving average (ARIMA).** ARIMA models are a powerful statistical tool used for time series analysis and forecasting [37]. These models are especially valuable when the data exhibit underlying patterns such as trends, seasonality, and autocorrelation. An ARIMA model consists of three main components: the autoregressive (AR) component captures the influence of past observations on the current value, the integrated (I) component addresses non-stationarity in the data by applying differencing to stabilize the mean, and the Moving Average (MA) component reflects the relationship between the current observation and past forecast errors. The structure of an ARIMA model is defined by three parameters—p, d, and q—which correspond to the orders of the AR, I, and MA components, respectively [38–41]. The entire model $ARIMA\ (p,\ d,\ q)$ can be specified as [42]:

$$\left(1 - \varphi_1 B - \ldots - \varphi_p B^p\right)\left(1 - B\right)^d y_t = \left(1 + \theta_1 B + \ldots + \theta_q B^q\right)\epsilon_t,\tag{2}$$

where $\{\varepsilon_t\} \sim WN\left(0, \sigma^2\right)$. A seasonal ARIMA known as SARIMA model defined as $ARIMA(p, d, q) \times (P, D, Q)^s$ process with seasonal period $s$,

$$\phi(B)\phi\left(B^s\right)\left(1 - B\right)^d\left(1 - B^s\right)^D y_t = \theta(B)\,\Theta\left(B^s\right)\varepsilon_t,\tag{3}$$

where $\{\varepsilon_t\} \sim WN\left(0, \sigma^2\right)$, and the polynomial $\phi(B)$, $\Phi\left(B^s\right)$, $\theta(B)$, and $\Theta\left(B^s\right)$ are defined as $\left(1 - \phi_1 B - \ldots - \phi_p B^p\right)$, $\left(1 - \Phi_1 B^s - \ldots - \Phi_p B^{sP}\right)$, $\left(1 + \theta_1 B + \ldots + \theta_p B^p\right)$, and $(1 + \Theta_1 B^s + \ldots + \Theta_Q B^{sQ})$, respectively.

**2.4.2. Trigonometric seasonality, Box-Cox transformation, ARMA errors, trend, and seasonal components (TBATS).** The TBATS model is a state-of-the-art time series forecasting method that effectively handles complex trends and multiple seasonal components [43]. To address non-linearity and non-normality in the data, the model applies the Box-Cox transformation, which enhances its ability to capture underlying trends and seasonal structures more accurately [44,45]. After extracting these components, the model fits an ARMA process to the residuals, to account for any remaining autocorrelation. This additional step helps to de-correlate the series and improve the accuracy of forecasts for future periods [46].

**2.4.3. Exponential smoothing state space (ETS).** The ETS model provides a versatile and dependable method for forecasting univariate time series, especially when the data show trends and/or seasonal variations. This model decomposes the series into three fundamental components: error, trend, and seasonality. Each of these components can be specified as additive (A), multiplicative (M), none (N), or automatic (Z), with the latter allowing the model to select the most appropriate form based on the data [47]. For example, an ETS (A, M, N) model applies an additive form for the error term, a multiplicative form for the trend, and assumes no seasonal variation. The flexibility of the ETS framework in accommodating various structural features makes it especially appropriate for numerous time series forecasting tasks. [48].

**2.4.4. Generalized autoregressive conditional heteroskedasticity (GARCH).** The GARCH model, which was first introduced by Bollerslev [49], is commonly utilized to analyze and predict time series data that exhibit changing volatility. This model comprises two elements: the autoregressive conditional heteroskedasticity (ARCH) component, which addresses the variance of the error term, and the GARCH component, which accounts for the relationship between the current and previous error variances. GARCH models are especially proficient at illustrating volatility clustering and long-memory effects within time series data. Nonetheless, they can be computationally demanding and typically need substantial datasets for accurate estimation [50].

**2.4.5. Artificial neural network (ANN).** Artificial Neural Networks (ANNs) are adaptable, nonlinear modeling systems that excel at identifying intricate relationships within data, which makes them particularly useful for time series forecasting

applications [51]. Numerous ANN architectures have been designed and effectively utilized in various fields, such as climate science, finance, and engineering [52]. Among these architectures, the multilayer perceptron (MLP) featuring a single hidden layer is one of the most widely employed structures for time series forecasting [53]. This configuration generally consists of an input layer, a hidden layer, and an output layer, with the connections between the nodes organized as feedforward, acyclic links [52].

**2.4.6. Facebook prophet (FP).** Prophet is a powerful forecasting model for time series, created by Facebook's Core Data Science team, intended to effectively manage data that shows significant trends and various seasonal influences [54,55]. It breaks down the time series into four components utilizing the additive model [56]:

$$y(t) = g(t) + s(t) + h(t) + \epsilon_t , \tag{4}$$

where $y(t)$ is the predicted value; $g(t)$ captures the trend (linear or logistic growth); and $s(t)$ accounts for seasonal effects (daily, weekly, yearly, etc.); $h(t)$ models the holiday-related irregularities; and $\in_t$ represents the error term. Although holiday effects were not considered relevant in this context, Prophet was selected for its strength in modeling long-term trends and complex seasonal structures. The model has been widely used in environmental and climatological applications for its flexibility, interpretability, and computational efficiency in handling interannual variation, long-term seasonality, and anomaly detection [57].

**2.4.7. Support vector regression (SVR).** SVR is an effective method for forecasting time series, based on the principles of Support Vector Machines (SVM) [58]. It functions by transforming input features into a higher-dimensional space using kernel functions, which allows it to capture complex dependencies that typical linear models might overlook [59]. This characteristic makes SVR especially well-suited for forecasting tasks that involve non-stationary and nonlinear time series. Additionally, SVR is robust against outliers and has demonstrated the ability to deliver consistent and dependable forecasts across a range of applications [60].

**2.4.8. Decision tree regression (DTR).** DTR is a supervised learning model that does not rely on parametric assumptions and is used to forecast continuous outcomes by systematically dividing the data into subsets based on feature values. At every internal node within the tree, a decision rule is employed to partition the data in a manner that reduces prediction error, usually by utilizing metrics like mean squared error. This process continues until a defined stopping condition is met, resulting in a tree structure where each leaf node represents a predicted numerical outcome [61]. This method can effectively capture nonlinear relationships and interactions among variables without needing data normalization or assumptions about the underlying distribution. While single decision trees may be susceptible to overfitting, their clarity and computational efficiency make them valuable for foundational forecasting in time series analysis [62].

**2.4.9. Random forest regression (RFR).** RFR is an ensemble learning technique that builds several decision trees during the training phase and averages their outputs to perform regression tasks [63]. This technique reduces the likelihood of overfitting often seen with single decision trees by combining predictions from trees that are formed using various bootstrap samples and subsets of features [64]. It enhances model stability and prediction accuracy, especially in cases with nonlinear relationships, outliers, or high-dimensional datasets. Moreover, RF models are robust against noise and require limited parameter adjustments, without making strong assumptions regarding the underlying distribution of the data [65,66]. Due to these attributes, RFR has been extensively utilized in time series prediction, environmental modeling, and other fields that require dependable forecasting capabilities.

**2.4.10. Extreme gradient boosting (XGBoost).** The XGBoost algorithm is an advanced ensemble learning method that enhances gradient boosting. It combines the predictions of multiple weak learners to achieve greater predictive accuracy than individual models. This technique is widely utilized in various scientific and engineering fields [67]. It is a decision tree-based ensemble machine learning approach that is extensively utilized in data science because of its high degree of accuracy and efficiency. By adopting a gradient boosting framework that consolidates the predictions from

multiple constituent trees, the model enhances performance incrementally and generates precise forecasts [68]. The XGBoost model uses a gradient descent-based optimization method to reduce the loss function, leading to increased prediction accuracy through a process of iterative model improvement [69]. Given its capability to effectively manage nonlinear relationships, feature interactions, and large datasets, XGBoost is particularly effective for complex forecasting challenges, such as time series predictions.

**2.4.11. Long short-term memory (LSTM).** LSTM networks are a specialized form of recurrent neural network (RNN) designed to handle sequential and time-dependent data [70]. They utilize memory cells along with gating mechanisms to effectively capture both short-term and long-term relationships, overcoming the vanishing gradient issue often faced by conventional RNNs. This makes LSTM models especially adept at time series forecasting tasks characterized by nonlinearity and temporal dependencies. LSTM networks feature gating mechanisms—including input, output, and forget gates—that allow the model to selectively keep, update, or eliminate information, enabling it to identify important temporal patterns in time series data [29,71].

**2.4.12. Gated recurrent unit (GRU).** GRUs are a type of recurrent neural network designed to model sequential data. They are a simplified version of LSTM networks and can effectively learn long-term patterns in time series. GRUs use gating mechanisms to decide what information to keep or discard, helping the model focus on important trends over time. Compared to LSTMs, GRUs are faster to train and require fewer computational resources, while still offering similar prediction accuracy. These features make them well-suited for time series forecasting tasks [72]. In practice, GRU models involve several hyperparameters that can influence prediction accuracy, including the number of hidden layers, batch size, and learning rate schedule. The number of hidden layers defines the model's depth, batch size determines the frequency of weight updates, and the learning rate schedule (e.g., learning rate drop) adjusts how the learning rate changes over time [73,74].

## 2.5. Hyperparameter tuning for ML time series models

The hyperparameters for all ML and deep learning models were optimized using Time Series Cross-Validation (TSCV) with grid search methodology [75]. This approach ensures robust parameter selection while maintaining the temporal integrity of the time series data. The tuning process employed rolling window cross-validation to prevent data leakage and ensure generalizability. For classical time-series models (ARIMA, ETS, TBATS, GARCH, etc.), automatic optimization techniques were utilized to tune the models through Akaike information criterion (AIC) minimization. ML models underwent extensive TSCV with grid search optimization. The SVR model was fine-tuned with a polynomial degree of 3, cost parameter of 45.69, nu value of 0.5, and tight tolerance and epsilon values of 0.00001 to ensure precise convergence. The RFR model was optimized with 100 trees, balancing computational efficiency with predictive performance. XGBoost parameters were carefully calibrated with "reg:squarederror" objective function, RMSE evaluation metric, learning rate of 0.1, maximum tree depth of 6, and subsample ratios of 0.8 for both instances and features, trained over 100 rounds with 10-round early stopping.

Deep learning architectures received specialized attention in their hyperparameter configuration. The LSTM network was structured with two hidden layers of 50 units each using ReLU activation, trained with Adam optimizer for 10 epochs with batch size of 30 and early stopping patience of 5 epochs. The GRU model employed a larger architecture with 100 units in both layers, similarly trained with Adam optimizer for 10 epochs with batch size of 30. Both recurrent neural networks utilized a consistent lookback period of 30 time steps to capture temporal dependencies effectively. The Prophet model maintained its default parameter settings, leveraging its inherent optimization capabilities for time series forecasting and the ANN model's network structure was automatically determined by the nnetar() function. This comprehensive hyperparameter tuning strategy ensured that each model was optimally configured for the specific characteristics of the time series data, balancing model complexity with generalization capability while respecting the chronological dependencies inherent in the dataset.

## 2.6. Seasonal-trend decomposition using loess (STL decomposition)

In time series analysis, decomposition techniques are essential tools for isolating underlying components such as trend, seasonality, and irregular fluctuations from observed data. One such widely used method is the STL, introduced by Cleveland et al. (1990) [76]. Unlike classical decomposition methods such as SEATS (Seasonal Extraction in ARIMA Time Series) and X-11, STL is highly flexible and capable of handling a wide range of seasonal patterns, including daily, sub-daily, monthly, and quarterly data [56,77]. This adaptability makes STL particularly suitable for complex or non-standard seasonal structures often found in climatological and environmental time series.

## 2.7. Model performance evaluation metrics

The model's accuracy was assessed using the following performance metrics: Mean Absolute Error (MAE), Mean Absolute Percentage Error (MAPE), Mean Absolute Scaled Error (MASE), and Root Mean Squared Error (RMSE). The formulas for calculating each of these metrics are provided below:

$$\text{MAE} = \frac{1}{n}\sum_{i=1}^{n}\left|y_i - \hat{y}_i\right|$$

(5)

$$\text{MAPE} = \frac{1}{n}\sum_{i=1}^{n}\frac{\left|y_i - \hat{y}_i\right|}{y_i} \times 100\%$$

(6)

$$\text{MASE} = \frac{1}{n}\sum_{i=1}^{n}\left(\frac{\left|y_i - \hat{y}_i\right|}{\frac{1}{n-m}\sum_{j=m+1}^{n}\left|y_j - y_{j-m}\right|}\right)$$

(7)

$$\text{RMSE} = \sqrt{\frac{1}{n}\sum_{i=1}^{n}\left(y_i - \hat{y}_i\right)^2}$$

(8)

where $n$ represents the total number of data points, $y_i$ refers to the actual observed values, and $\hat{y}_i$ denotes the predicted values. The symbol $m$ represents the seasonal period. MAE, RMSE, MAPE, and MASE are common metrics used to evaluate the accuracy of forecasting models. MAE and RMSE are affected by the scale of the data, while MAPE and MASE are scale-independent.

## 2.8. Ethical statement

Ethical approval was not required for this study, as the dataset used does not involve any human or animal subjects.

## 3. Results

### 3.1. Data description

Rajshahi, situated in northwestern Bangladesh, is among the hottest regions in the country. To assess thermal discomfort in this heat-prone area, we utilized daily average air temperature (°C) and relative humidity (%) data from the Bangladesh Meteorological Department (BMD), spanning January 1985 to December 2024. The DI was computed using Thom's formula, offering a standardized indicator of heat-related stress. Out of 14,610 days analyzed, 4,692 were classified as "High Discomfort," where most of the population would feel discomfort. Additionally, 806 days fell under the "Severe

Discomfort" category, suggesting widespread stress. Notably, no days met the criteria for "Extreme Heat Stress," implying that although discomfort is common, life-threatening thermal extremes were not observed during this period.

Fig 3 illustrates the daily changes in the DI for Rajshahi from January 1985 to December 2024. The blue spikes represent the daily DI values, which follow a consistent seasonal pattern, rising sharply during the warmer months and declining in the cooler periods each year. This pattern emphasizes the regular cycles of thermal discomfort experienced in the region. The maroon line represents the seasonal-adjusted values, which smooth out the predictable fluctuations caused by the changing seasons. This adjustment allows us to observe short-term variations and irregularities that may result from unusual weather or broader climate changes. Notably, the trend line (gray) shows no significant upward or downward movement, indicating that there have been no abrupt long-term changes in DI during the 40 years.

Fig 4 displays the annual distribution of the DI in Rajshahi from 1985 to 2024. While Fig 3 indicates consistent seasonal patterns, Fig 4 shows a gradual increase in the median DI values over time. Most years after 2005 have median values consistently above 25. This implies that while the overall pattern of seasonal discomfort remains stable, the intensity of heat stress has progressively increased, leading to more frequent extreme discomfort events in recent decades. This long-term trend aligns with the broader warming patterns observed in the region.

Fig 5 illustrates the monthly distribution of the DI in Rajshahi throughout the entire study period. Each boxplot represents the range of daily DI values for every month, highlighting the seasonal fluctuations in human thermal discomfort over the year. The findings indicate a clear seasonal trend, with discomfort levels gradually rising from January to June, peaking between June and September, and then decreasing as the cooler months of November and December approach. During the summer and early autumn months (June to September), the DI remains consistently elevated, with median values hovering around or surpassing 27, signifying an extended period of increased thermal stress. Furthermore, these months display narrower interquartile ranges, implying that the higher discomfort levels are more stable and persistent during this period. Conversely, the winter and early spring months (December to March) are characterized by lower DI values and greater variability, reflecting cooler and more variable conditions.

Fig 6 complements this pattern by showing the monthly distribution of discomfort categories based on the discomfort index. Throughout the winter months (December to February), almost every day is categorized as "No discomfort," reflecting thermally pleasant conditions. As temperatures begin to rise in March, levels of discomfort also increase, resulting in

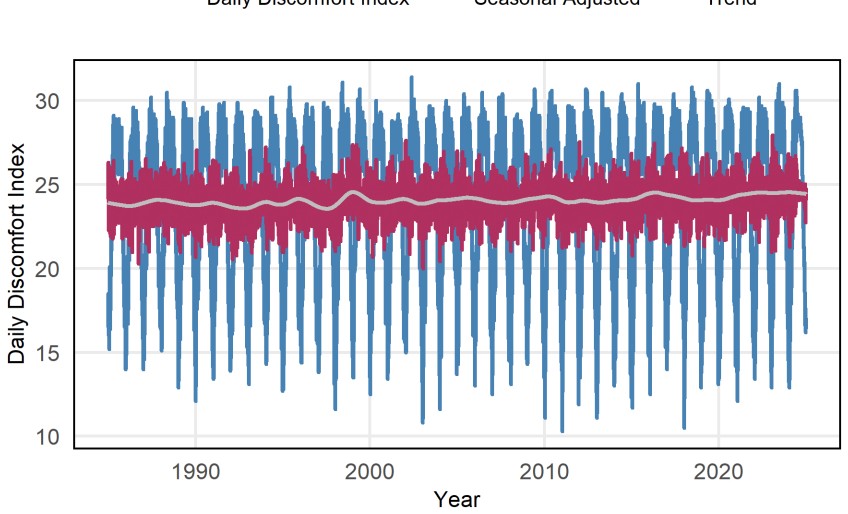

**Fig 3. Daily discomfort index in Rajshahi with seasonal adjustment (maroon) and trend components (gray) from January 1985 to December 2024.**

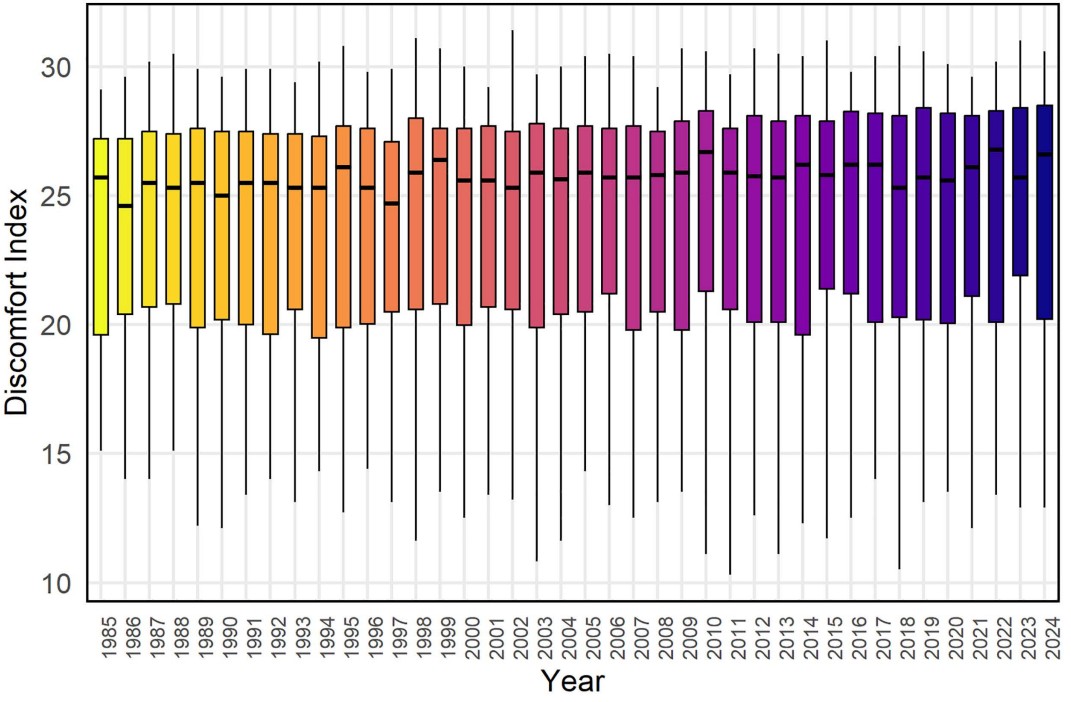

**Fig 4.** Boxplot shows the distribution of yearly thermal discomfort variation in Rajshahi from January 1985 to December 2024.

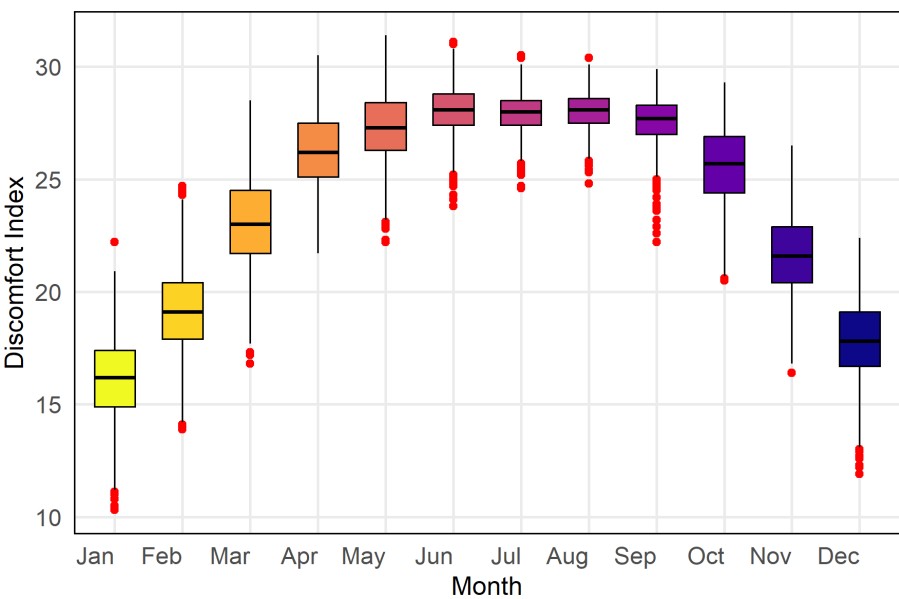

**Fig 5.** Boxplot shows the distribution of monthly thermal discomfort variation in Rajshahi from January 1985 to December 2024.

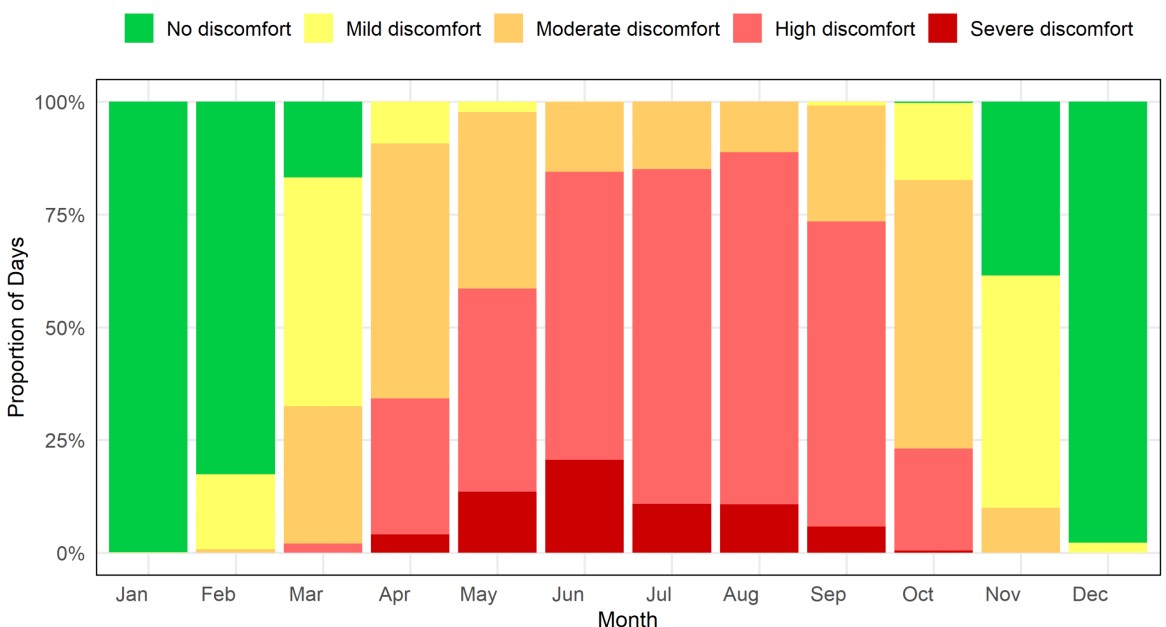

**Fig 6. Monthly proportion of population discomfort levels in Rajshahi based on daily Discomfort Index from January 1985 to December 2024, categorized by defined thermal stress thresholds.**

a higher percentage of days classified as "Mild discomfort" and "Moderate discomfort." From April through September, thermal stress intensifies, especially between June and August, where the majority of days are categorized as "High discomfort" and there is also a notable number of "Severe discomfort" days. Interestingly, there are no days rated as "Extreme Heat Stress," signifying that such conditions are absent. In October, the trend starts to change, with a reduction in high discomfort levels, leading to a return to more comfortable conditions by November. This seasonal cycle illustrates a transition from winter thermal comfort to widespread discomfort in the summer months.

Fig 7 summarizes these seasonal dynamics by plotting the monthly average DI values against established comfort thresholds. The lowest DI values are recorded in January (~16.1) and December (~17.8), both within the no-discomfort zone. Starting in March, DI levels rise sharply, reaching 23.0 (Mild Discomfort), and remaining above 26 from April to September. The peak occurs in June (DI ≈ 28.1), falling within the "High Discomfort" category. No monthly average exceeds the extreme stress threshold, which aligns with the findings from previous figures. Overall, this pattern highlights extended periods of moderate to high discomfort during the pre-monsoon and monsoon months.

### 3.2. Forecasting Discomfort Index (DI) without seasonal adjustment

To forecast the DI without adjusting for seasonality in Rajshahi, a city known for its intense heat in northwestern Bangladesh, we examined high-frequency time series data spanning from 1985 to 2024. Due to the non-stationary and volatile characteristics of DI—especially without seasonal decomposition—accurate forecasting presents significant challenges. To address this issue, we conducted an extensive evaluation of over 76 model configurations, including traditional time series models (ARIMA, ETS, TBATS, GARCH), eight standalone ML models (e.g., ANN, SVR, XGBoost, LSTM, GRU), and a diverse array of hybrid combinations – including ARIMA-based hybrids (e.g., ARIMA-SVR), TBATS-based hybrids, ETS-based hybrids, GARCH-based hybrids, and cross-hybrids such as SVR-ARIMA, LSTM-TBATS, etc. A complete listing of all hybrid models without seasonal adjustment and their performance metrics is provided in Table S1 in S1 File.

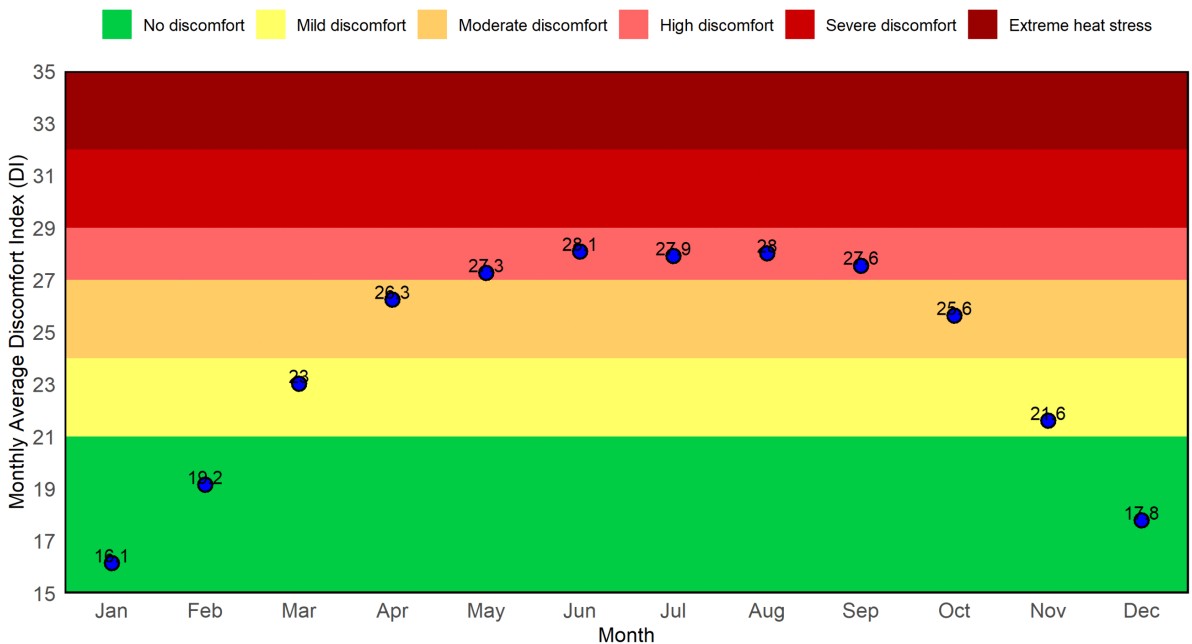

**Fig 7. Monthly average Discomfort Index (DI) overlaid on comfort zones in Rajshahi from January 1985 to December 2024, illustrating typical thermal stress levels experienced in each month.**

The forecasting performance of the standalone models and the top five best performing hybrid models for daily DI without seasonal adjustment in Rajshahi is presented in Table 2. Among the standalone models, tree-based algorithms demonstrated superior capability in handling the raw, high-frequency data. XGBoost emerged as the top performer (MAE: 0.7132, MAPE: 3.13%), slightly outperforming SVR and Random Forest (RFR), while traditional models like ARIMA and ETS performed poorly (MAPE > 18%), highlighting their limitations with non-stationary series. The hybrid model evaluation revealed that combinations leveraging the decomposition strength of TBATS with a subsequent machine learning regressor were most effective (Table 2). The top-performing models in this category were TBATS-DTR and TBATS-LSTM, which achieved the lowest errors (e.g., MAE < 0.68, MAPE ≈ 2.95%). This superior performance underscores the advantage of first using TBATS to model the series' inherent trend and seasonality, then applying a non-linear model (DTR or LSTM) to capture complex patterns in the residuals (Table 2). In contrast, hybrids that lacked a coherent decomposition strategy or combined incompatible components (e.g., GARCH-GRU, LSTM-TBATS) exhibited significantly higher error rates, confirming that a strategic integration of models is critical for success (Table S1 in S1 File). To enhance predictive accuracy, we implemented a seasonal-adjusted forecasting framework using STL decomposition. This approach isolates the trend, seasonal, and residual components, allowing models to be applied to the adjusted series. We developed a comprehensive suite of seasonal-adjusted single and hybrid models, with full results of seasonal-adjusted hybrid models available in Table S2 in S1 File.

Among the single models applied to the STL-adjusted data in Table 3, STL-TBATS and STL-GRU were the most accurate, achieving low MAE values of 0.7957 and 0.8100, respectively. This indicates that even after STL decomposition, components like complex seasonality and non-linear residuals persist, which these models are well-suited to handle. The seasonal-adjusted hybrid models demonstrated a further significant improvement in performance. The top-performing model overall was the STL-TBATS-LSTM hybrid in Tables 3 and S2 in S1 File, which achieved the lowest scores across all metrics (MAE: 0.4809, MAPE: 2.12%). The STL-TBATS-DTR model was a close competitor. The success of the

**Table 2. Forecasting performance of single models and the top five hybrid models for daily Discomfort Index (DI) without seasonal adjustment in Rajshahi (1985–2024).**

| Model | MAE | MAPE | RMSE | MASE |
|---|---|---|---|---|
| ARIMA | 4.0303 | 18.3358 | 4.5762 | 5.5674 |
| TBATS | 1.0676 | 4.7198 | 1.3481 | 1.4748 |
| ETS | 5.4390 | 21.6258 | 6.1355 | 7.5134 |
| GARCH | 5.3192 | 21.2736 | 5.9868 | 7.3478 |
| ANN | 3.6676 | 17.0398 | 4.8501 | 5.0663 |
| FP | 1.1337 | 4.9328 | 1.4040 | 1.5661 |
| SVR | 0.7135 | 3.1399 | 0.9529 | 0.9857 |
| RFR | 0.7226 | 3.1591 | 0.9431 | 0.9982 |
| DTR | 1.0899 | 4.6834 | 1.3389 | 1.5056 |
| XGBoost | 0.7132 | 3.1321 | 0.9389 | 0.9852 |
| LSTM | 3.9641 | 18.6616 | 4.6702 | 5.4759 |
| GRU | 3.9683 | 18.4550 | 4.5826 | 5.4817 |
| ARIMA-DTR | 0.7082 | 3.1184 | 0.9447 | 0.9782 |
| ARIMA-LSTM | 0.7088 | 3.1198 | 0.9451 | 0.9791 |
| TBATS-DTR | 0.6709 | 2.9508 | 0.8940 | 0.9268 |
| TBATS-LSTM | 0.6712 | 2.9515 | 0.8946 | 0.9272 |
| XGBoost-TBATS | 0.7108 | 3.1263 | 0.9378 | 0.9819 |

**Table 3. Forecasting performance of seasonal-adjusted single models and the top five seasonal-adjusted hybrid models for daily Discomfort Index (DI) in Rajshahi (1985–2024).**

| Model | MAE | MAPE | RMSE | MASE |
|---|---|---|---|---|
| STL-ARIMA | 2.0336 | 8.9547 | 2.2265 | 2.8092 |
| STL-TBATS | 0.7957 | 3.4746 | 1.0112 | 1.0991 |
| STL-ETS | 1.8099 | 7.9998 | 2.0173 | 2.5002 |
| STL-GARCH | 7.4974 | 32.2394 | 7.5532 | 10.3568 |
| STL-ANN | 1.7015 | 7.2473 | 2.0123 | 2.3504 |
| STL-FP | 0.8680 | 3.7492 | 1.0855 | 1.1990 |
| STL-SVR | 3.8460 | 17.8282 | 4.3918 | 5.3128 |
| STL-RFR | 3.8259 | 17.7987 | 4.3787 | 5.2850 |
| STL-DTR | 3.5909 | 16.8797 | 4.1825 | 4.9604 |
| STL-XGBoost | 3.8372 | 17.8228 | 4.3891 | 5.3007 |
| STL-LSTM | 1.0124 | 4.5584 | 1.2768 | 1.3986 |
| STL-GRU | 0.8100 | 3.5280 | 1.0265 | 1.1189 |
| STL-TBATS-RFR | 0.4877 | 2.1510 | 0.6456 | 0.6737 |
| STL-TBATS-DTR | 0.4811 | 2.1217 | 0.6378 | 0.6646 |
| STL-TBATS-LSTM | 0.4810 | 2.1230 | 0.6381 | 0.6644 |
| STL-TBATS-GRU | 0.4828 | 2.1251 | 0.6389 | 0.6669 |
| STL-ARIMA-RFR | 0.4975 | 2.1909 | 0.6628 | 0.6872 |

STL-TBATS-LSTM architecture can be attributed to its complementary, multi-stage design: the STL decomposition first removes the dominant seasonality; TBATS then expertly models any remaining complex seasonal and trend components in the adjusted series; finally, the LSTM captures the intricate non-linear and long-term temporal dependencies present in the residuals of the TBATS model. This layered approach effectively addresses both linear and non-linear patterns,

making it exceptionally powerful for forecasting the DI. In summary, while several models showed competence, the strategic hybrid combination of STL, TBATS, and LSTM proved to be the most accurate and reliable method for forecasting thermal discomfort in Rajshahi. This model's architecture is particularly adept at handling the complex, multi-component nature of the DI time series.

Fig 8 shows the daily DI in Rajshahi, both observed (1985–2024) and forecasted (2025–2027), categorized by thermal stress levels. Fig 8a presents the long-term DI trends, while the bottom panel zooms in on the forecast period to highlight seasonal discomfort patterns. Historically, DI values generally stayed within the "No Discomfort" to "Moderate Discomfort" range, indicating relatively comfortable conditions. However, the forecast indicates a significant increase in thermal stress. Fig 8b reveals frequent spikes into the "High Discomfort" and "Severe Discomfort" zones, especially during the pre-monsoon and summer months. This suggests a growing climate risk, with more days likely to surpass critical heat thresholds, posing challenges to health and well-being.

Fig 9, the donut chart, summarizes this trend quantitatively by showing the percentage distribution of forecasted days across categorized discomfort levels. A large portion of days (39.8%) are predicted to reach "High Discomfort" ($27 \leq DI < 29$), meaning most people will experience hot and uncomfortable conditions. This is followed by "No Discomfort" (26%), "Moderate Discomfort" (20.3%), and "Mild Discomfort" (12.7%). Notably, "Severe Discomfort" ($29 \leq DI < 32$) is expected on 1.2% of days, indicating significant heat stress for nearly everyone. Although no days are projected to reach the "Extreme Heat Stress" threshold ($DI \geq 32$), the increasing frequency of days in the higher discomfort categories points to a concerning upward trend. Combining Figs 8 and 9, we observed that demonstrate a clear intensification of heat-related discomfort in Rajshahi's near future. This pattern calls for proactive climate adaptation strategies, including improved heat warning systems, urban greening, and targeted health interventions to protect vulnerable populations.

## 4. Discussion

Globally, climate change is intensifying heat-related stress, especially in tropical and low-income countries [78]. Thermal discomfort, measured by indicators such as the DI, serves as a more reliable measure of heat-related stress that better represents human experience than temperature alone [79]. As rising temperatures and humidity levels interact with changing climatic regimes, DI is becoming an important tool for assessing public health risk [80]. In Bangladesh, particularly in northwestern regions such as Rajshahi, growing evidence indicates a rise in both the frequency and intensity of thermally uncomfortable days [22]. This pattern is especially pronounced during the pre-monsoon and monsoon seasons, indicating that there is an immediate need for forecasting technologies to aid climate-sensitive planning and safeguard susceptible communities.

This study proposed and evaluated 64 ML-based hybrid models for forecasting the Discomfort Index (DI) in Rajshahi, a heat-prone region of Bangladesh. To enhance forecasting accuracy, these combinations were also analyzed with seasonal adjustment using STL decomposition, resulting in a total of 128 model variations. Among the non-seasonally adjusted models, the TBATS-DTR hybrid yielded the best performance, with an MAE of 0.6709, MAPE of 2.9508, RMSE of 0.8940, and MASE of 0.9268. The TBATS-LSTM model followed closely, achieving nearly identical metrics, indicating its high effectiveness. After applying STL decomposition, the STL-TBATS-LSTM model achieved the highest overall accuracy (MAE = 0.4810, MAPE = 2.1230, RMSE = 0.6381, MASE = 0.6644), with the STL-TBATS-DTR model showing comparable results. These findings confirm that STL-based decomposition significantly enhances model performance and that hybridization, particularly involving LSTM, consistently improves forecasting accuracy. Furthermore, the study observed a notable increase in the frequency and intensity of thermally uncomfortable days, particularly during the pre-monsoon and monsoon seasons. This trend, supported by both historical data and model projections, signals an escalating thermal discomfort burden in Rajshahi. As shown in Fig 8, while historical data mostly fell within the "No Discomfort" to "Moderate Discomfort" categories, future forecasts for 2025–2027 indicate a significant shift toward more extreme conditions. Specifically, "High Discomfort" days are projected to dominate, with 39.8% forecasted days, while "Severe Discomfort" is

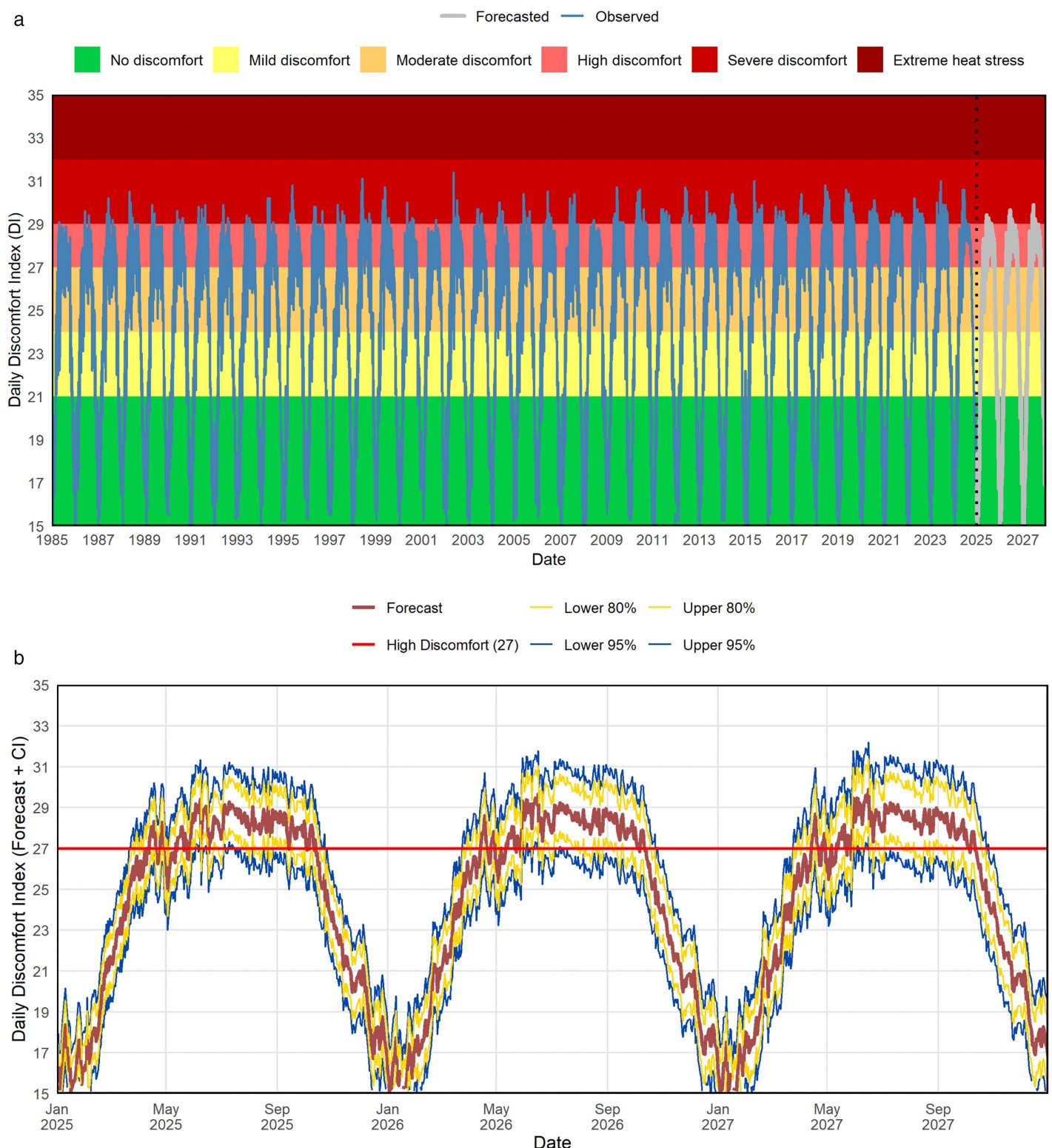

**Fig 8. Observed (1985–2024) and forecasted (2025–2027) daily Discomfort Index (DI) in Rajshahi using the STL-TBATS-LSTM model, overlaid with categorized thermal stress levels in (a) and the forecasted DI (2025–2027) with 80% and 95% confidence intervals to highlight the uncertainty ranges with seasonal and extreme discomfort patterns in (b).**

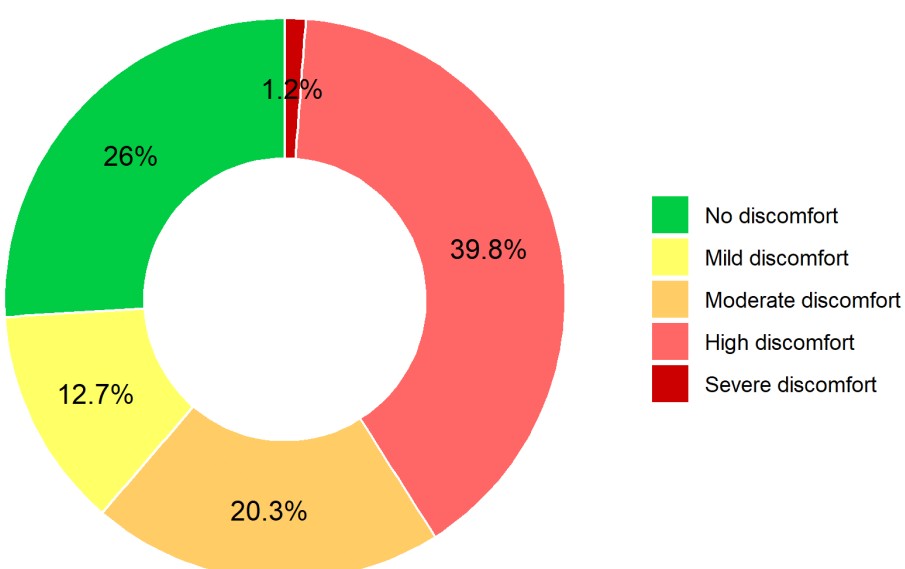

**Fig 9. Donut chart illustrates percentage distribution of forecasted daily Discomfort Index (DI) categories in Rajshahi from January 2025 to December 2027.**

predicted for 1.2% days (Fig 9). These projections align with recent findings on the pronounced impact of seasonal variation on health outcomes, particularly for vulnerable populations [81], underscoring that accurate seasonal forecasting is critical for managing these escalating heat-related health risks.

The forecasted increase in "High Discomfort" days has profound implications for public health and urban management. To translate these model outputs into actionable strategies, stakeholders can use these forecasts for targeted planning. For urban and health authorities in Bangladesh, we recommend the following specific policy and adaptation strategies: (i) Enhanced Early Warning Systems, (ii) Adaptive Work Scheduling, and (iii) Informed Infrastructure Design. (i) Enhanced Early Warning Systems: Integrate the STL-TBATS-LSTM model forecasts into a city-level heat-health alert system. Timely DI-based warnings can trigger public health advisories, directing citizens to cooling centers and initiating hydration programs, especially for the elderly, children, and those with pre-existing conditions who are most susceptible to seasonal health shocks [81,82]. This study's model can play a crucial role in shaping early warning systems, allowing public health authorities to issue timely alerts ahead of discomfort peaks. (ii) Adaptive Work Scheduling: Labor and industry departments can use seasonal forecasts to mandate adjusted work schedules during peak discomfort periods, reducing heat exposure for outdoor workers in construction and agriculture. In practice, these forecasts can inform the establishment of cooling centers, hydration stations, targeted outreach programs, and work/rest schedules for vulnerable populations [83]. Furthermore, (iii) Informed Infrastructure Design: Urban planners should use long-term discomfort projections to guide investments in heat mitigation infrastructure. This includes prioritizing green spaces, cool roofs, and improved building insulation in areas projected to experience the most severe discomfort, thereby mitigating the urban heat island effect [84]. By embedding these forecasts into operational frameworks, authorities can proactively protect public health and enhance urban resilience.

The findings of this study align closely with prior research on thermal discomfort, climate variability, and hybrid forecasting models. The increasing frequency and intensity of thermally uncomfortable days in Rajshahi reflect regional trends identified by Ekra et al. (2024) and Rahman et al. (2021), who reported rising heat stress indicators across Bangladesh, particularly in the coastal and northern regions [19,22]. Similarly, Talukdar et al. (2017) and Sharmin et al. (2020)

emphasized the seasonal prevalence of discomfort in Mymensingh and Dhaka, respectively, highlighting the nationwide impact of climate-induced thermal stress [20,21]. Our findings further support global projections made by Zhang et al. (2023) and Ullah et al. (2024), which indicate that low-income tropical regions are expected to experience increased thermal discomfort due to climate change. [15,16].

Our results emphasize the effectiveness of hybrid forecasting models that combine ML with classical time series methods. The improved performance of the STL-TBATS-LSTM model in this study aligns with the findings of Qureshi et al. (2025), who reported that their STL-ARIMA-LSTM hybrid model outperformed others in predicting heatwaves in Bangladesh [29]. Both studies demonstrate the importance of incorporating seasonal decomposition (STL) to capture patterns in high-frequency environmental data more effectively. Similarly, Saif et al. (2024) found that a hybrid model combining ARIMA and neural networks achieved the highest forecasting accuracy for fish production in the country [30]. Ferdoush et al. (2021) also showed that a hybrid model integrating Random Forest with bidirectional LSTM outperformed standalone LSTM models in short-term electricity load forecasting in Bangladesh [32]. Collectively, these studies highlight the growing success of hybrid approaches in addressing complex forecasting challenges across various domains in the region. Furthermore, our findings are consistent with earlier international research by Susanti et al. (2017), Sulandari et al. (2020), and Yunis et al. (2024), which demonstrated that hybrid models that integrate statistical and machine learning techniques provide higher forecasting accuracy across a range of fields, including air pollution, electricity load, and hydrology [25,26,28].

This study presents several key strengths that enhance the credibility and applicability of its findings. First, it employs a high-frequency dataset consisting of daily observations over 40 years (1985–2024). This extensive temporal resolution allows for an accurate capture of both short-term fluctuations and long-term trends in thermal discomfort, ensuring greater statistical reliability and sensitivity to seasonal and interannual variability. Second, the study systematically evaluates a diverse array of hybrid model combinations (128 in total), encompassing both traditional and machine learning approaches. By integrating models such as ARIMA, TBATS, and ETS with advanced learners like DTR, RFR, and LSTM, it thoroughly investigates the forecasting potential of both linear and nonlinear structures. Third, the consistent outperformance of STL-adjusted hybrid models—especially those utilizing LSTM—demonstrates the practical advantages of combining statistical rigor with deep learning capabilities. This strategy not only improves prediction accuracy but also enhances the model's adaptability to complex and evolving patterns in climate data. Finally, the study goes beyond model evaluation; it interprets the results in terms of human impact. By linking forecasted discomfort levels to public health risks and seasonal exposure patterns, it provides actionable insights for urban planners and policymakers in heat-prone regions such as Rajshahi.

The proposed models demonstrated strong performance, but several limitations should be acknowledged. First, the DI was calculated using only air temperature and relative humidity, excluding other influential variables like wind speed and solar radiation due to data unavailability. Second, although the models showed excellent accuracy for Rajshahi, their applicability to other regions may be limited without further validation. Third, while STL decomposition improved forecast performance, it assumes a stable seasonal pattern over time, which may not fully capture nonlinear shifts driven by climate change [85,86]. External validation against independent datasets, such as ERA5 reanalysis or nearby weather stations, would further strengthen the robustness of the findings; however, this was beyond the scope of the current study and is recommended as an important direction for future research. Moreover, hybrid models that incorporate deep learning components like LSTM are computationally intensive, requiring longer training times and greater resources. This could limit their real-time operational use [87]. However, these limitations are somewhat mitigated by the use of high-frequency (daily) data collected over a robust 40-year period, which enhances the statistical reliability and temporal sensitivity of the models. Importantly, none of these limitations undermines the validity of the key findings, which remain sound and valuable for forecasting and planning in similar heat-prone contexts.

Recognizing these limitations creates opportunities for future work. First, future studies could explore additional decomposition methods like Empirical Mode Decomposition (EMD), robust STL, or wavelet transform [88] to capture more

complex patterns in climate data. Integrating these techniques with newer deep learning architectures, like Transformer models [89], may further enhance prediction accuracy. Second, while this study focused on the DIin Rajshahi, the methodology could be applied to other regions of Bangladesh or similar heat-prone areas worldwide. Comparative studies across different climatic zones would help evaluate the generalizability and adaptability of the proposed hybrid models. Third, incorporating additional meteorological and socioeconomic variables could refine DI calculations and enable more targeted public health strategies. Additionally, while this study primarily focused on error metrics for model evaluation, the incorporation of comprehensive temporal validation strategies and sensitivity analyses remains beyond the current scope and will be addressed in future research to further enhance model robustness and reliability. Finally, collaboration with health data systems could facilitate the development of early warning systems that connect DI forecasts with hospital admission trends, supporting real-time decision-making and climate-resilient urban planning in vulnerable regions like Rajshahi.

## 5. Conclusion

This study demonstrates that hybrid forecasting models integrating seasonal-trend decomposition (STL), advanced time series modeling (TBATS), and deep learning (LSTM) substantially improve the accuracy of thermal discomfort predictions in Rajshahi, Bangladesh. With climate change intensifying heat stress, our findings reveal a marked increase in the frequency and severity of discomfort days, underscoring a growing public health challenge. The STL-TBATS-LSTM model provides actionable forecasts that can inform heat-health early warning systems, adaptive labor policies, and climate-resilient urban planning, empowering stakeholders to mitigate heat-related risks effectively. While some limitations, such as variable selection and regional generalizability, remain, this work lays a critical foundation for enhancing climate adaptation strategies in heat-prone regions through robust, interpretable, and operational forecasting tools.

## Supporting information

**S1 File. Supplementary tables showing forecasting performance of hybrid models for daily Discomfort Index (DI) with and without seasonal adjustment in Rajshahi (1985–2024).**
(DOCX)

**S1 Data. Dataset of Discomfort Index (1985–2024).**
(XLSX)

## Acknowledgments

The authors are grateful to the editor and the anonymous reviewers for their constructive comments and suggestions, which have substantially strengthened the quality of this manuscript. The authors also acknowledge the Climate Division of the Bangladesh Meteorological Department (BMD) for generously providing the data utilized in this analysis.

## Author contributions

**Conceptualization:** Mohammad Ashraful Haque Mollah, Rumana Rois.

**Data curation:** Amrin Binte Ahmed, Md. Mahin Uddin Qureshi, Mohammad Mahboob Hussain Khan.

**Formal analysis:** Amrin Binte Ahmed, Md. Mahin Uddin Qureshi, Adisha Dulmini.

**Investigation:** Amrin Binte Ahmed, Mohammad Mahboob Hussain Khan, Rumana Rois.

**Methodology:** Amrin Binte Ahmed, Md. Mahin Uddin Qureshi.

**Software:** Amrin Binte Ahmed, Mohammad Mahboob Hussain Khan.

**Supervision:** Rumana Rois.

**Visualization:** Amrin Binte Ahmed, Mohammad Mahboob Hussain Khan.

**Writing – original draft:** Amrin Binte Ahmed, Md. Mahin Uddin Qureshi, Adisha Dulmini.

**Writing – review & editing:** Amrin Binte Ahmed, Rumana Rois.

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
