## [Decision Letter · Decision Letter 0]

18 Aug 2025

Dear Dr. Binte Ahmed,

Thank you for submitting your manuscript to PLOS ONE. After careful consideration, we feel that it has merit but does not fully meet PLOS ONE’s publication criteria as it currently stands. Therefore, we invite you to submit a revised version of the manuscript that addresses the points raised during the review process.

We look forward to receiving your revised manuscript.

Kind regards,

Dost Muhammad Khan, PhD

Academic Editor

PLOS ONE

Journal Requirements:

4. In this instance it seems there may be acceptable restrictions in place that prevent the public sharing of your minimal data. However, in line with our goal of ensuring long-term data availability to all interested researchers, PLOS’ Data Policy states that authors cannot be the sole named individuals responsible for ensuring data access (http://journals.plos.org/plosone/s/data-availability#loc-acceptable-data-sharing-methods).

We require you to either (1) present written permission from the copyright holder to publish these figures specifically under the CC BY 4.0 license, or (2) remove the figures from your submission

Reviewers' comments:

Reviewer's Responses to Questions

**Comments to the Author**

1. Is the manuscript technically sound, and do the data support the conclusions?

Reviewer #1: Yes

Reviewer #2: Yes

2. Has the statistical analysis been performed appropriately and rigorously?

Reviewer #1: Yes

Reviewer #2: Yes

3. Have the authors made all data underlying the findings in their manuscript fully available?

Reviewer #1: No

Reviewer #2: Yes

4. Is the manuscript presented in an intelligible fashion and written in standard English?

Reviewer #1: Yes

Reviewer #2: Yes

Reviewer #1: Dear editor, thank you for the invitation to review this article, my comments are listed below.

1. The introduction should more explicitly clarify the novelty of combining STL, TBATS, and LSTM in the context of forecasting thermal discomfort in Bangladesh. While the methodology is strong, the practical advantages of this hybrid combination compared to prior studies or conventional models should be more clearly positioned, especially regarding implications for public health interventions and urban planning.

2. The model performance section is exhaustive but can be made more reader-friendly. Consider summarizing key findings by focusing on the top 3 to 5 performing models rather than detailing extensive metrics for all combinations. A concise comparative statement about why certain hybrid models outperformed others would improve readability without sacrificing analytical depth.

3. The discussion could better connect forecasted increases in discomfort levels to specific policy recommendations or adaptation strategies for Bangladesh’s urban and health authorities. Adding a paragraph that suggests how stakeholders could use these forecasts for planning, such as early warning systems, work scheduling, or infrastructure design, would elevate the applied value of the research.

4. Consider adding more recent literature that examines the impact of seasonal variation on health outcomes, particularly for vulnerable populations (https://doi.org/10.1016/j.ijtst.2024.11.009), to enhance the relevance and contextual grounding of your work.

Reviewer #2: The manuscript presents a valuable contribution by combining hybrid forecasting models to evaluate thermal discomfort in Rajshahi, Bangladesh, within the context of climate change, but several areas require improvement to enhance clarity, reproducibility, and applicability. While the paper is generally well-written and logically structured, some technical sections particularly the hybrid model descriptions should be simplified for interdisciplinary readers, and minor editorial refinements would improve readability. Figures are referenced but not fully integrated, requiring consistent formatting, proper resolution, and clear labeling, while acronyms such as TBATS, ETS, and GRU should be briefly redefined upon first use for non-specialist readers. The literature review is comprehensive, yet the discussion could better connect findings to public health, climate adaptation, and urban planning strategies. The focus on Rajshahi offers depth but limits generalizability, and the methodology’s adaptability to other regions should be briefly discussed. Although the study evaluates 128 hybrid models, it lacks theoretical justification for why STL-TBATS-LSTM was expected to perform best, and critical details on hyperparameter tuning are missing, undermining reproducibility. Validation could be strengthened by adding temporal strategies like walk-forward validation or sensitivity analysis, and incorporating uncertainty estimates or prediction intervals would improve decision-making relevance. While the seasonal discomfort trends align with regional climate patterns, external validation with datasets such as ERA5 would enhance credibility. The discussion of limitations, including the assumption of stationary climate trends and potential future mitigation effects, is insufficient and should be expanded. Finally, the forecasts highlight thermal risks but could offer stronger recommendations for public health preparedness and urban planning. Addressing these issues particularly data transparency, model justification, reproducibility, and uncertainty communication would significantly improve the manuscript’s rigor, accessibility, and impact.

**Do you want your identity to be public for this peer review?** For information about this choice, including consent withdrawal, please see our Privacy Policy

Reviewer #1: No

Reviewer #2: No

---

## [Author Response · Author response to Decision Letter 1]

4 Nov 2025

October 26, 2025

Manuscript ID: PONE-D-25-28698

Title: Application of Seasonal-Adjusted Hybrid Models for Forecasting Discomfort Index in a Heat-Prone Region of Bangladesh

Journal: PLOS ONE

Dear Editor,

We sincerely thank you and the reviewers for your valuable time and constructive comments on our manuscript. We are grateful for the opportunity to revise and resubmit. We have carefully addressed each concern, and the manuscript has been revised accordingly. Below we provide a detailed point-by-point response.

Best regards,

Authors

Editorial Requirements

Author’s Response:

Thank you for this reminder. We have carefully reviewed the provided PLOS ONE style templates and confirm that our manuscript now fully adheres to all formatting requirements, including file naming conventions, reference style, and manuscript structure.

Author’s Response:

We will share all author-generated code without restrictions in a public GitHub repository https://github.com/amrinbinteahmed/Discomfort-Index-Data-and-Code/tree/main upon manuscript acceptance. The repository will be cited in the manuscript, include a detailed README file. We confirm adherence to PLOS ONE's code sharing guidelines.

Author’s Response:

We have uploaded the fully anonymized dataset as a Supporting Information file accompanying this manuscript. We confirm our adherence to PLOS ONE's open data policy. The Data Availability statement has been updated accordingly.

4. In this instance it seems there may be acceptable restrictions in place that prevent the public sharing of your minimal data. However, in line with our goal of ensuring long-term data availability to all interested researchers, PLOS’ Data Policy states that authors cannot be the sole named individuals responsible for ensuring data access (http://journals.plos.org/plosone/s/data-availability#loc-acceptable-data-sharing-methods).

Author’s Response:

We have uploaded the fully anonymized dataset as a Supporting Information file accompanying this manuscript.

We require you to either (1) present written permission from the copyright holder to publish these figures specifically under the CC BY 4.0 license, or (2) remove the figures from your submission

Author’s Response:

We thank the editor for this important note. We confirm that Figure 1 is an original creation by the authors and does not contain any copyrighted material. The map was created by the authors using the open-source software QGIS (version 3.30), utilizing our study data. The figure caption has been updated to include the required attribution: "The map was created using QGIS Geographic Information System, version 3.30 's-Hertogenbosch (https://qgis.org/project/visual-changelogs/visualchangelog330/)." We hereby grant permission for it to be published under the CC BY 4.0 license.

Author’s Response:

Thank you. We have followed these instructions.

Reviewer #1 Comments

Comment 1: The introduction should more explicitly clarify the novelty of combining STL, TBATS, and LSTM in the context of forecasting thermal discomfort in Bangladesh. While the methodology is strong, the practical advantages of this hybrid combination compared to prior studies or conventional models should be more clearly positioned, especially regarding implications for public health interventions and urban planning.

Author’s Response:

Thank you for your insightful comment regarding the novelty and positioning of our hybrid model combining STL, TBATS, and LSTM for forecasting thermal discomfort in Bangladesh. We have updated the introduction to explicitly highlight these novel contributions and their public health and urban planning implications, as follows "The novelty of our method lies in the strategic integration of seasonal-trend decomposition using Loess (STL) for effective seasonal decomposition, an advanced time series model (like Trigonometric Seasonality, Box-Cox Transformation, ARMA Errors, Trend, and Seasonal Components (TBATS)) for managing complex and multiple seasonal patterns, and an ML model (like LSTM) to capture nonlinear temporal dynamics. This integrated hybrid model surpasses conventional methods by delivering enhanced accuracy in forecasting thermal discomfort, specifically tailored to Bangladesh’s highly variable climate. The resulting high-precision projections are not only a technical advancement but also play a crucial role in enabling proactive public health interventions, optimizing early warning systems, and guiding data-driven urban planning. Such improvements are essential for strengthening climate resilience among vulnerable populations in heat-prone regions like Rajshahi.”

Comment 2: The model performance section is exhaustive but can be made more reader-friendly. Consider summarizing key findings by focusing on the top 3 to 5 performing models rather than detailing extensive metrics for all combinations. A concise comparative statement about why certain hybrid models outperformed others would improve readability without sacrificing analytical depth.

Author’s Response:

Thank you for such insightful comments. We have streamlined the model performance section to focus on the top-performing models. For non-seasonally adjusted data, XGBoost, SVR, and RFR were the best among standalone models, whereas TBATS-DTR and TBATS-LSTM led hybrid models. For seasonally adjusted data, STL-TBATS-LSTM emerged as the top hybrid model, closely followed by STL-TBATS-DTR. These models benefit from combining decomposition (STL, TBATS) with nonlinear learning (LSTM, decision trees), effectively capturing both linear and nonlinear temporal and seasonal structures, supporting their superior forecasting accuracy. A complete listing of all hybrid models with or without seasonal adjustment and their performance metrics is provided in Supplementary Tables S1 and S2.

Comment 3: The discussion could better connect forecasted increases in discomfort levels to specific policy recommendations or adaptation strategies for Bangladesh’s urban and health authorities. Adding a paragraph that suggests how stakeholders could use these forecasts for planning, such as early warning systems, work scheduling, or infrastructure design, would elevate the applied value of the research.

Author’s Response:

We have expanded the Discussion section to provide actionable recommendations. We emphasize how forecasts can inform early warning systems, heat mitigation policies, urban planning interventions, and health advisories targeted at vulnerable populations. This contextualizes model outputs within practical climate adaptation and public health frameworks relevant to Bangladesh. Thank you for this good suggestion.

Comment 4: Consider adding more recent literature that examines the impact of seasonal variation on health outcomes, particularly for vulnerable populations (https://doi.org/10.1016/j.ijtst.2024.11.009), to enhance the relevance and contextual grounding of your work.

Author’s Response:

We appreciate this suggestion and have incorporated the referenced recent study on seasonal variation and health outcomes, highlighting its relevance to our work. This strengthens the literature context in the Discussion section and supports the importance of accurate seasonal forecasting in managing heat-related health risks.

Reviewer #2 Comments

1. Basic Reporting

(i) Language and Clarity: The manuscript is generally well-written and structured, but some technical sections (particularly the hybrid model descriptions) could be simplified for interdisciplinary readers. Minor editorial improvements are needed to enhance readability.

Author’s Response:

We simplified technical sections, particularly hybrid model descriptions, and made minor editorial improvements to enhance readability.

(ii) Figures and Visuals: Several figures are referenced but not fully integrated (e.g., “Insert Figure 1”). Final versions of all visuals should be included with consistent formatting, clear labeling, and appropriate resolution. Complex multi-panel visuals may benefit from division into sub-figures or supplementary materials.

Author’s Response:

All figures have been finalized with consistent formatting, clear labels, and high resolution. As per PLOS ONE’s submission guidelines, figures have been uploaded separately and will be automatically integrated into the reviewer PDF.

(iii) Terminology and Abbreviations: While core concepts such as STL, DI, and hybrid models are introduced, some model acronyms (e.g., TBATS, ETS, GRU) would benefit from brief redefinitions upon first use to aid non-specialist readers.

Author’s Response:

All acronyms like TBATS, ETS, GRU have been redefined upon first use to aid interdisciplinary readers.

(iv) Contextual Relevance: The literature review is comprehensive and well-situated in the context of climate change and thermal discomfort. However, the implications for public health and urban planning could be emphasized more strongly in the discussion section.

Author’s Response:

The Discussion now more strongly emphasizes public health and urban planning implications.

(v) Data Availability: The Data Availability Statement mentions restricted access. Clarifying how other researc

---

## [Decision Letter · Decision Letter 1]

23 Feb 2026

Application of Seasonal-Adjusted Hybrid Models for Forecasting Discomfort Index in a Heat-Prone Region of Bangladesh

PONE-D-25-28698R1

Dear Authors,

We’re pleased to inform you that your manuscript has been judged scientifically suitable for publication and will be formally accepted for publication once it meets all outstanding technical requirements.

Kind regards,

Anurag Barthwal, Ph.D.

Academic Editor

PLOS One

Additional Editor Comments (optional):

Dear Authors,

Your manuscript has now completed the second round of peer review. In the second round, both reviewers confirmed that their concerns have been satisfactorily addressed and recommended acceptance.

I am therefore pleased to inform you that your manuscript is accepted for publication in its current form. Congratulations, and thank you for choosing our journal for the dissemination of your work.

Kind regards,

Dr Anurag Barthwal

Academic Editor

Reviewers' comments:

Reviewer's Responses to Questions

**Comments to the Author**

Reviewer #1: (No Response)

Reviewer #2: All comments have been addressed

2. Is the manuscript technically sound, and do the data support the conclusions?

Reviewer #1: (No Response)

Reviewer #2: Yes

3. Has the statistical analysis been performed appropriately and rigorously?

Reviewer #1: (No Response)

Reviewer #2: Yes

4. Have the authors made all data underlying the findings in their manuscript fully available?

Reviewer #1: (No Response)

Reviewer #2: Yes

5. Is the manuscript presented in an intelligible fashion and written in standard English?

Reviewer #1: (No Response)

Reviewer #2: Yes

Reviewer #1: (No Response)

Reviewer #2: The revised manuscript titled “Application of Seasonal-Adjusted Hybrid Models for Forecasting Discomfort Index in a Heat-Prone Region of Bangladesh” has addressed all the reviewer’s comments satisfactorily. The authors have improved the clarity of the technical sections, properly finalized and integrated all figures, and clearly explained the rationale for selecting the STL-TBATS-LSTM hybrid model. The discussion section has been strengthened to better highlight the public health and urban planning implications, and relevant recent literature has been added. Data availability concerns have been resolved through the upload of an anonymized dataset, and limitations regarding generalizability, validation, and climate assumptions have been appropriately acknowledged. Overall, the manuscript is well-written, methodologically sound, and makes a meaningful contribution to forecasting thermal discomfort and supporting climate adaptation planning. I therefore recommend acceptance for publication.

**Do you want your identity to be public for this peer review?** For information about this choice, including consent withdrawal, please see our Privacy Policy

Reviewer #1: No

Reviewer #2: No

---

## [Editor Report · Acceptance letter]

PONE-D-25-28698R1

PLOS One

Dear Dr. Binte Ahmed,

I'm pleased to inform you that your manuscript has been deemed suitable for publication in PLOS One. Congratulations! Your manuscript is now being handed over to our production team.

Kind regards,

on behalf of

Dr. Anurag Barthwal

Academic Editor

PLOS One